# A Unique In Vitro Assay to Investigate ABCB4 Transport Function

**DOI:** 10.3390/ijms24054459

**Published:** 2023-02-24

**Authors:** Csilla Temesszentandrási-Ambrus, Gábor Nagy, Annamária Bui, Zsuzsanna Gáborik

**Affiliations:** 1SOLVO Biotechnology, Charles River Laboratories Hungary, H-1117 Budapest, Hungary; 2Doctoral School of Molecular Medicine, Semmelweis University, Tűzoltó u. 37-47, H-1094 Budapest, Hungary

**Keywords:** ABCB4 inhibitors, MDR3, hepatotoxicity, drug-induced liver injury

## Abstract

ABCB4 is almost exclusively expressed in the liver, where it plays an essential role in bile formation by transporting phospholipids into the bile. ABCB4 polymorphisms and deficiencies in humans are associated with a wide spectrum of hepatobiliary disorders, attesting to its crucial physiological function. Inhibition of ABCB4 by drugs may lead to cholestasis and drug-induced liver injury (DILI), although compared with other drug transporters, there are only a few identified substrates and inhibitors of ABCB4. Since ABCB4 shares up to 76% identity and 86% similarity in the amino acid sequence with ABCB1, also known to have common drug substrates and inhibitors, we aimed to develop an ABCB4 expressing Abcb1-knockout MDCKII cell line for transcellular transport assays. This in vitro system allows the screening of ABCB4-specific drug substrates and inhibitors independently of ABCB1 activity. Abcb1KO-MDCKII-ABCB4 cells constitute a reproducible, conclusive, and easy to use assay to study drug interactions with digoxin as a substrate. Screening a set of drugs with different DILI outcomes proved that this assay is applicable to test ABCB4 inhibitory potency. Our results are consistent with prior findings concerning hepatotoxicity causality and provide new insights for identifying drugs as potential ABCB4 inhibitors and substrates.

## 1. Introduction

Multidrug resistance protein 3 (MDR3, ABCB4) is predominantly expressed in the canalicular membrane of hepatocytes. It has been predicted to be a floppase that translocates phosphatidylcholine (PC) from the inner to the outer leaflet of the membrane bilayer [1,2,3]. Translocated PCs are available for extraction by bile salts mixed micelles, and exert two essential functions in the bile: they protect the biliary tree from the detergent activity of bile salts [4,5], and maintain cholesterol solubility, preventing supersaturation with cholesterol [6].

In agreement with these proposed roles, ABCB4 mutations result in a broad spectrum of phenotypes ranging from progressive familial intrahepatic cholestasis type 3 (PFIC3) to ABCB4-related cholestatic liver disorders of varying manifestation and severity in adults [7]. Clinical studies revealed the strong association between ABCB4 gene mutations and low-phospholipid-associated cholelithiasis syndrome (LPAC), a rare type of gallstone disease [8,9].

Historically, ABCB4 was classified as multidrug resistance transport protein based on 76% identity and 86% similarity of its amino acid sequence to the highly promiscuous ABCB1 transporter [10]. In a recent publication on cryo-EM studies with chimeric transporters of ABCB1 and ABCB4, a shared overall transport mechanism has been proposed. It suggests that ABCB4 transports PC to the outer membrane leaflet by an alternative access mechanism with the entire phospholipid molecule entering and leaving a central translocation pathway, in contrast to mechanisms proposed for other phospholipid transporters [11]. However, the loss of ABCB4 function is not readily compensated by ABCB1 or by other mechanisms [12]. In mouse knockout studies Abcb1 is unable to compensate for loss of Abcb4 function, as these mice develop a liver disease that appears to be caused by the complete inability of the liver to secrete phospholipids into the bile [5]. In a similar fashion, stable transfectants of LLC-PK1 cells revealed that ABCB1 has broad specificity for phospholipids, but ABCB4 expressing cells exclusively released short chain phosphatidylcholine [13].

In contrast, ABCB4 can recognize and transport ABCB1 substrates, but its role in conferring drug resistance is still inconclusive. Although the transport rate was low for most ABCB1 substrates, Smith et al. demonstrated directional transport of digoxin, paclitaxel, vinblastine and ivermectin through ABCB4-transfected LLC-PK1 cells. Digoxin transport was inhibited by typical ABCB1 inhibitors such as verapamil, cyclosporin A or valspodar [5,13,14,15,16]. These results also suggest that ABCB4 is primarily a PC transporter that can translocate various typical ABCB1 substrates as well. Even though ABCB4 showed approximately 10-fold less ATPase activity than ABCB1, this activity is enough to confer multidrug resistance in chimera protein containing ABCB1 transmembrane domains [17]. The potential role of ABCB4 in multidrug resistance was indicated by increased ABCB4 transcript levels in paclitaxel-, doxorubicin-, and vincristine-resistant cell lines [18,19,20]. ABCB4 overexpression was also correlated with high-risk Wilms tumors [21].

Drug-induced liver injury (DILI) is a leading cause of drug development termination; therefore, its prediction in early stages is crucial [22,23]. Because of its role in the process of bile formation, inhibition of ABCB4 by drugs is believed to contribute to DILI and impact the hepatocellular/biliary toxicity of bile acids [24,25]. Drug-induced cholestasis (DIC), a frequent manifestation of DILI [26], is caused by alterations in the hepatobiliary disposition of bile acids, which in turn, is a result of direct injury to the bile ducts or inhibition of bile acid formation or transport [27,28]. Several drugs cause cholestasis by inhibiting canalicular efflux transporters [7,29]; however, there are significant differences in the relevance of biliary transporter inhibition in the development of DILI [30,31]. Inhibition of ABCB11 (Bile Salt Export Pump, BSEP) is a major risk factor in the development of cholestatic hepatotoxicity, though recent publications do not fully underscore this correlation [32,33]. To improve the predictive power, multidrug resistance proteins (MRPs) are included in screens with varying degrees of success [30,33,34]. In contrast to ABCB11, inhibition of ABCB4 in cholestatic diseases has received little attention to date.

Impaired ABCB4-mediated biliary phospholipid secretion was shown to be involved in itraconazole-induced cholestasis, where biliary PC levels were markedly reduced, while biliary bile salt levels remained unchanged [24]. While studying antifungal agents, Mahdi et al. described the inhibitory effect of ketoconazole and posaconazole in LLC-PK1-ABCB4 cells [35].

In a primary hepatocyte assay, ABCB4-mediated PC transport was investigated using several structurally distinct cholestatic drugs [25]. In a similar study, Aleo et al. tested 125 drugs grouped by their DILI potential, and identified several drugs which are shared inhibitors of both ABCB4 and ABCB11 [36]. These results suggest that such inhibitors may exacerbate the cholestatic effect [35].

Although in vitro ABCB4 assays have been already reported, only a few addressed the role of ABCB4 in drug interactions [25,36]. Hence, compared with other drug transporters, ABCB4 has a small number of identified drug substrates and inhibitors. Therefore, we aimed to investigate ABCB4 activity and interaction with drugs independently of other transporters in an in vitro assay.

Here, we developed an Abcb1 knockout (Abcb1KO) MDCKII-ABCB4 cell line, which shows a polarized morphology and completely lacks Abcb1 background activity but has ABCB4 activity. Using this cell line, the interaction of several hepatotoxic, anticancer as well as ABCB1 interactor drugs with ABCB4-mediated digoxin transport was investigated. We showed that this assay system is also well suited to identify potential drugs highly specific for ABCB4 by excluding overlapping specificity from Abcb1.

## 2. Results

### 2.1. Characterization of the Abcb1 Knockout MDCKII Cell Line Engineered to Express ABCB4

An Abcb1 biallelic knockout MDCKII (Abcb1KO-MDCKII) cell clone was generated by GenScript (Leiden, the Netherlands) using the CRISPR/Cas9 system. To verify biallelic KO, the canine Abcb1-related properties of the Abcb1KO-MDCKII and MDCKII parental cells were characterized. The transcript levels of Abcb1 in the cell lines were analyzed by RT-qPCR (Appendix A). The parental MDCKII and Abcb1KO-MDCKII cells were examined by FACS using a calcein AM assay. Compared with the MDCKII parental cells, the Abcb1KO-MDCKII cells exhibited a high fluorescent signal, corresponding to the lack of canine Abcb1 efflux activity (Appendix A).

To confirm the impact of endogenous canine Abcb1 on the efflux ratio (ER) on two prototypic Abcb1 substrates, bidirectional transport experiments with digoxin and talinolol (both 1 µM) were performed. In the MDCKII parental cells, apparent permeability (Papp) in the B-A direction was higher than in the A-B direction, with an average ER of 4.74 ± 0.15 for digoxin and 2.70 ± 0.57 for talinolol, indicating active transport by canine Abcb1. In contrast, digoxin and talinolol Papp were identical in both directions in Abcb1KO-MDCKII cells, with an ER close to unity (Figure 1a). The Abcb1 inhibitor valspodar decreased the efflux ratios of both digoxin and talinolol close to unity in the MDCKII parental cells, whereas in the Abcb1KO-MDCKII cells the ER remained unaffected. In a previous study, using this Abcb1KO-MDCKII cell line, we also confirmed that maraviroc has a strong affinity for endogenous canine Abcb1 in MDCKII cell lines, despite the lack of transport in the Abcb1KO-MDCKII cell line [37].

For the purpose of this study, we expressed the ABCB4 protein in this Abcb1KO-MDCKII cell line. Expression of human ABCB4 mRNA were determined in the Abcb1KO-MDCKII-Mock and Abcb1KO-MDCKII-ABCB4 cells by RT-qPCR (Appendix A). The ABCB4 mRNA level (determined by the formula 2^−(ΔΔCT)^) was approximately 6000-fold higher in the Abcb1KO-MDCKII-ABCB4 cells than in Abcb1KO-MDCKII-Mock cells. Canine Abcb1 mRNA could not be detected in any of the cell lines.

Protein expression of human ABCB1 and ABCB4 were assessed by Western blot in the Abcb1KO-MDCKII-ABCB4, Abcb1KO-MDCKII-Mock and previously established Abcb1KO-MDCKII-ABCB1 cells (from now on referred as ABCB4, Mock and ABCB1 cells, respectively). Human ABCB1 protein was detected in the ABCB1 cells, but not in the ABCB4 cells, and ABCB4 was expressed only in the ABCB4 cell line. Neither ABCB1 nor ABCB4 was detectable in the Mock cells (Appendix A).

### 2.2. Characterization of ABCB4 Transporter Function in the Abcb1KO-MDCKII-ABCB4 Cell Line in Comparison with ABCB1 Function in Abcb1KO-MDCKII-ABCB1 Cells

Since the substrate specificity of ABCB4 and ABCB1 was reported to be overlapping, we tested five known ABCB1 substrates (quinidine, prazosin, fexofenadine, digoxin, and ketoconazole at 1 µM final concentration) in bidirectional transport assays using ABCB1, Mock and ABCB4 cell lines (Figure 1b). While digoxin is a shared substrate for ABCB1 and ABCB4, it shows higher ER in the ABCB1-expressing cell line compared with ABCB4. The other compounds span a wide range of ERs (3.47 to 39) in the ABCB1-expressing cells. In contrast, quinidine and fexofenadine showed ER < 2 in the ABCB4 cells, which is a cut-off value for active efflux. In the case of prazosin, minor ABCB4-mediated directional transport was detected. Ketoconazole is an ABCB1 inhibitor [38] and an ABCB1 substrate [39]. Accordingly, an ER of 5.54 for ketoconazole was seen in the ABCB1 cell line. Under the experimental conditions of the present study, ketoconazole was also identified as an ABCB4 substrate (ER 2.78). The corresponding ERs in the control experiments using Mock cells were at, or very close to, unity.

For further functional validation of the assay, digoxin was used as an ABCB4 probe substrate. ABCB4-mediated digoxin transport experiments were conducted as a time course by taking samples from both compartments at various time points (1, 2, 3, 4 and 6 h). During the 6-h incubation, digoxin transport was roughly linear. Therefore, taking samples at one fixed time point should suffice for Papp calculations, and the 3 h time point was chosen for this (Appendix A). The transepithelial transport of digoxin was measured for 3 h using a range of initial concentrations (Appendix A); however, because of the low aqueous solubility and potentially higher Km, the kinetic parameters of digoxin transport could not be determined.

### 2.3. Screening ABCB4 Interactors with Different DILI Concern to Test the Predictive Potential of the ABCB4 Transport Assay Using Digoxin

To investigate the predictive nature of the ABCB4 bidirectional assay in potential drug-ABCB4 interactions, inhibition studies were performed using 30 drugs with different DILI outcomes (18 Most-DILI concern, 9 Less-DILI concern and 3 No-DILI concern) from the U.S. Food and Drug Administration (FDA) Liver Toxicity Knowledge Base (LTKB). In addition, seven structurally diverse compounds were also tested in ABCB4 inhibition assays. The concentration-dependent effect of the selected compounds on ABCB4-mediated digoxin (1 µM) transport were determined.

The known ABCB4 inhibitors itraconazole [24,25,35,40] and verapamil [16,25,36] showed potent inhibition of ABCB4 with estimated IC_50_ values of 0.17 and 0.39 µM, respectively. Ketoconazole (IC_50_ 0.56 µM), ritonavir (IC_50_ 0.73 µM), saquinavir (IC_50_ 1.4 µM) and valspodar (IC_50_ 0.15 µM) also inhibited ABCB4-mediated digoxin transport. As shown in Table 1, fluconazole, amiodarone, carbamazepine, minoxidil, furosemide and acetylsalicylic acid have no effect on ABCB4-mediated transport of digoxin. The results are in agreement with the previous studies on hepatocytes [25,36]. We also identified ivermectin as an ABCB4 inhibitor in our assay, which is not surprising since it was described earlier that ABCB4 can bind and transport ivermectin in LLC-PK1-ABCB4 cells [16]. We did not detect interaction with methotrexate in our assay, in contrast to previous data [36]. In the case of benzbromarone, we observed the opposite phenomenon compared with other reported inhibitors. Instead of an IC_50_ value of 0.4 µM [36], a significantly higher IC_50_ value (19.89 µM) was determined in the ABCB4 cells.

Among drugs with Less-DILI-concern, the ABCB1 modulator amlodipine [41] showed a weak interaction with the ABCB4 protein. Fenofibrate, felodipine and pantoprazole showed no interaction with the ABCB4 transporter at the concentrations we tested, whereas carvedilol proved to be a potent inhibitor with an IC_50_ value of 0.70 µM.

Known ABCB1-interacting compounds were also investigated on the ABCB4 cells. We found that valspodar [16] is a potent ABCB4 inhibitor as previously described, and zosuquidar, elacridar, mibefradil also elicited a strong concentration-dependent inhibitory effect. Quinidine, a weak ABCB4 substrate, potently inhibited ABCB4 function with an IC_50_ value of 1.09 µM. Prazosin also inhibited ABCB4 with an IC_50_ value of 16.12 µM.

The tyrosine kinase inhibitors (TKIs) gefitinib, imatinib, sorafenib and erlotinib are classified as Most-DILI-concern drugs, and idelalisib can cause clinically apparent liver injury [42]. In our assay, the most potent ABCB4 inhibitors were gefitinib and imatinib (IC_50_ 0.81 and 1.24 µM), whereas sorafenib and erlotinib showed less potent inhibition (IC_50_ 4.42 and >5µM). Idelalisib did not inhibit ABCB4.

We also identified antiviral agents (lopinavir, darunavir and asunaprevir) as ABCB4 inhibitors, which has not previously been described in the literature. Representative inhibition curves are shown in Appendix A.

**Table 1 ijms-24-04459-t001:** Tested compounds with ABCB4 inhibition activities and with ABCB1 interaction abilities.

DILI Concern	Compound	IC_50_ (µM), Based on Digoxin ER	In Vitro Literature Data (IC_50_, µM)	ABCB1 Interactor
Most DILI concern	gefitinib	0.81	^a^	Y [43]
	imatinib	1.24	^a^	Y [44]
	sorafenib	4.42	^a^	Y [45]
	erlotinib	>5 µM, 60% inhibition at 5 µM	^a^	Y [46]
	fluconazole	>3, NI	>300 [36]	N [38]
	itraconazole	0.17	2.1 [25], 30% inhibition at 10 µM [35], 50% inhibition at 1 µM [24], 22.5 µM [40]	Y [38]
	ketoconazole	0.56	5.6 [25], 4.6 [40]	Y [38]
	lopinavir	0.6	^a^	Y [47]
	ritonavir	0.73	9.6 [25], 11.3 [36]	Y [47]
	darunavir	>10 µM, 60% inhibition at 10 µM	^a^	Y [48]
	amiodarone	>10, NI	>300 [36], >100 [40]	Y [49]
	benzbromarone	19.89	0.4 [36]	Y (in house data)
	carbamazepine	>80, NI	>300 [36]	Conflicting information [50]
	clarithromycin	>80, NI	^a^	Y [51]
	methotrexate	>10, NI	3.1 [36]	Y [52]
	levofloxacin	>80, NI	^a^	Y [53]
	diltiazem	48.61	^a^	Y [54]
	cyclosporin A	0.46	>100 ^b^ [55], inhibitor ^d^ [16]	Y [56]
Less DILI concern	saquinavir	1.4	12.9 [25]	Y [57]
	fenofibrate	>20, NI	^a^	Y [58]
	ivermectin	0.53	Substrate [16]	Y [47]
	amlodipine	17.86	^a^	Y [41]
	pantoprazole	>10, NI	^a^	Y [59]
	felodipine	>30, NI	^a^	Y [60]
	carvedilol	0.70	^a^	Y [61]
	quinidine	1.09	^a^	Y [62]
	verapamil	0.39	6.3 [25], 7 [36], inhibitor ^d^ [16]	Y [63]
No DILI concern	minoxidil	>20, NI	>300 [36]	N [64]
	furosemide	>80, NI	>100 [25]	Conflicting information [65]
	acetylsalicylic acid	>100, NI	>300 [36]	N [66]
other	idelalisib	>5, NI	^a^	Y [67]
	asunaprevir ^c^	>5, NI	^a^	Y [68]
	valspodar	0.15	Inhibitor ^d^ [16]	Y [69]
	prazosin ^e^	16.12	^a^	Y [70]
	zosuquidar	0.07	^a^	Y [71]
	mibefradil	0.40	^a^	Y [72]
	elacridar	0.15	^a^	Y [73]

NI—no inhibition observed, ^a^ There are no data available, ^b^ 30–50% inhibition at concentrations ranging from 1–100 µM with no clear dose response, ^c^ Asunaprevir is not curated in the LTKB database, ^d^ IC_50_ value was not determined, ^e^ Ambiguous DILI-concern.

### 2.4. Identification of a Novel ABCB4 Transporter Substrate

As a next step, we also tested whether the two most potent inhibitors in the group of TKIs are ABCB4 substrates. Gefitinib at a concentration of 0.5 µM showed directional transport with an ER higher than 2 (4.02 at 60 min, 2.54 at 120 min and 3.23 at 180 min) in the ABCB4 cell line. The corresponding ERs in the Mock cells were very close to unity (Figure 2a). ABCB4-mediated transport of gefitinib was inhibited by itraconazole (2 µM). These results indicate that gefitinib is an inhibitor and a substrate of human ABCB4. In contrast, the corresponding ERs in ABCB4 and Mock cells were below 2 for imatinib, suggesting that although imatinib inhibits the function of ABCB4, it is not a substrate for this transporter (Figure 2b).

## 3. Discussion

Finding an efficient non-hepatocyte based in vitro model to study interactions between chemical entities and ABCB4 is still challenging. A few ABCB4 non-hepatocyte-based cellular assays have been reported, including ABCB4-transfected LLC-PK1 and HEK293 cells [13,74,75]. However, a major disadvantage of these cell lines is that they retain the expression of endogenous transport proteins [76,77,78], which might functionally interfere with the introduced transporter of interest [79]. Previous studies support that cell lines with a genomic knockout of the endogenous transporters, e.g., canine Abcb1, are more sensitive and overall, more suitable for assays of an introduced transporter [80,81].

In the present study, we successfully developed a cell line, Abcb1KO-MDCKII-ABCB4, which lacks endogenous canine Abcb1 activity and expresses the human ABCB4 transporter. Thorough characterization of the cell line with digoxin, a shared substrate of both ABCB1 and ABCB4, resulted in an assay capable of detecting ABCB4 inhibitors and substrates in an easy-to-use format. This cell line was then utilized to test a versatile array of drugs as potential inhibitors and/or substrates of ABCB4. For these inhibition studies, 30 drugs with different DILI outcomes were chosen from the Liver Toxicity Knowledge Base of the FDA, and seven more, structurally diverse, compounds were added to this list. In line with the literature data, we have here confirmed the interactions of many of these drugs (itraconazole, verapamil, ketoconazole, ritonavir, saquinavir and valspodar) with the ABCB4 protein in bidirectional assays. When comparing our results with those presented by He et al. [25] and Aleo et al. [36], however, a few things are worth noting. First, the above-mentioned primary hepatocyte assays utilize labeled PC to test ABCB4-dependent transport inhibition. He et al. tested the inhibitory potency of structurally distinct cholestatic drugs (chlorpromazine, imipramine, itraconazole, haloperidol, ketoconazole, saquinavir, clotrimazole, ritonavir, and troglitazone) [25]. In the latter study 125 drugs, grouped by the severity of liver injury caused, were tested [36]. Our assay detected all interactions reported in these studies, with only one exception, and drugs that did not interact with the ABCB4 protein in hepatocytes did not interact in our system either. Of note is that the IC_50_ values for ABCB4 inhibition obtained in our study are much lower than those obtained in hepatocyte assays. The results suggest that our assay can detect potential ABCB4 interactors with higher sensitivity compared with primary hepatocyte assays. Possible explanations include the difference in the substrate used and/or transporter protein abundance.

For a correct interpretation of MDCKII assay results, specific features of these cells need to be considered. For instance, the IC_50_ value of benzbromarone in our assay was much higher than in hepatocytes. Benzbromarone is metabolized by CYP2C9, and its toxic effect is believed to be mainly caused by its metabolites [82]. It can be speculated that the potent inhibitory effect observed in hepatocytes is partially caused by the metabolite. In general, in hepatocyte-based assays, both the parent drug and its metabolites may affect the assay readout [83], whereas in MDCKII cells, production of metabolites does not interfere with the effect of the parent drug.

Another feature of the MDCKII bidirectional assay is that transcellular transport as well as intracellular accumulation might be limited by the diffusion rate across the basolateral membrane or depend on the presence of endogenous SLC transporters. Such an example is methotrexate, which did not show inhibition in our assay. However, methotrexate potently inhibited ABCB4 with an IC_50_ of 3.1 µM in a human primary hepatocyte assay [36]. Since methotrexate is a hydrophilic dicarboxylic acid and shows low Papp in MDCKII cells [84], it is hypothesized that its transport is limited by the diffusion rate and cannot accumulate intracellularly to exert its inhibitory effect. In contrast, OATP1B1 and OATP1B3 can substantially contribute to the intracellular accumulation of methotrexate in hepatocytes [55,85].

Our results show that several potent ABCB1 inhibitors, such as verapamil, ketoconazole, valspodar, zosuquidar, cyclosporin A and mibefradil, inhibit ABCB4 activity. TKIs, such as gefitinib, imatinib, sorafenib and erlotinib, are classified as Most-DILI-concern drugs and are well known ABCB1 interactors [67,86,87,88,89]. In our assay, we demonstrated for the first time that all four TKIs also inhibited the function of ABCB4, which might contribute to the DILI properties of these drugs. Additionally, we identified gefitinib as a substrate of ABCB4. To our best knowledge, this is the first demonstration of the interaction of these TKIs with ABCB4.

Previous results suggest that several drugs are shared inhibitors of ABCB4 and ABCB11 transporters [36], supporting the earlier finding by Mahdi et al. that if a drug inhibits the function of both transporters, its cholestatic effect may increase [35]. These results imply that during drug development, combined drug–drug interaction tests, which include the transporters involved in hepatocellular bile acid homeostasis and bile formation (ABCB11, ABCB4 and ABCC2), may help to screen for drug candidates causing liver injury. However, the functional and clinical impacts of ABCB4 inhibition in the development of DILI require further investigation.

Unlike hepatocyte-based assays, our ABCB4 cell line is suitable to identify even low affinity non-phospholipid substrates of ABCB4. This feature of our assay is put into context by recent studies that point to the importance of ABCB4 in non-hepatobiliary tumors. High levels of ABCB4 expression have been reported in different leukaemias, even without co-expression with ABCB1 [90]. In leukaemia cells, daunorubicin accumulation is dependent on transporter inhibition with Cyclosporin A [91], indicating that ABCB4 is able to remove ABCB1 substrates from cancer cells and may play an active role in drug resistance. In a similar pattern, ABCB4 mediated the efflux transport of doxorubicin in vitro and contributed to the acquired resistance of doxorubicin in breast cancer cells [92]. Previous reports showed that ABCB4 and ABCC1 overexpression correlates with high-risk and significantly shorter disease-free survival time in patients with Wilms tumors (WT). This suggests the role of ABCB4 and ABCC1 in drug resistance observed in WT treatment [21].

In conclusion, we report here a novel assay that specifically measures ABCB4 activity using an ABCB4-overexpressing canine Abcb1 knockout MDCKII cell line. In terms of inhibition of this transporter, the positive and negative hits obtained in hepatocytes were reproduced in our assay with even higher sensitivity. We propose that the Abcb1KO-MDCKII-ABCB4 bidirectional assay could be used to screen the potential of chemical entities to inhibit ABCB4 transport activity to improve DILI prediction or assist in elucidating the mechanism of DIC. This is a superior system for ABCB4 substrate identification to aid drug development.

## 4. Materials and Methods

### 4.1. Materials

Non-radiolabeled chemicals were obtained from Merck/Sigma-Aldrich. Calcein AM was purchased from Invitrogen. All chemicals were of analytical grade. ^3^H-digoxin ([^3^H(G)], 23.8 Ci/mmol), ^3^H-prazosin ([7-methoxy-^3^H], 77.4 Ci/mmol) and Ultima Gold XR scintillation fluid were purchased from PerkinElmer (Waltham, MA, USA). ^3^H-quinidine ([9-^3^H], 20 Ci/mmol) and ^3^H-talinolol ([ring-^3^H(G)], 20 Ci/mmol) were purchased from American Radiolabeled Chemicals (St. Louis, MO, USA). ^3^H-fexofenadine ([^3^H], 4.9 Ci/mmol) was purchased from Moravek Biochemicals Inc. (Brea, CA, USA). The Tetro™ cDNA Synthesis Kit was purchased from Meridian Bioscience (London, UK). The Light Cycler^®^ 480 SYBR Green 1 master kit was obtained from Roche Applied Science (Foster City, CA, USA). 4–15% precast polyacrylamide gel (Invitrogen, Carlsbad, CA, USA) and ProSieve^TM^ QuadColor^TM^ Protein Marker (4.6–300 kDa) were obtained from Lonza (Basel, Switzerland). Monoclonal human ABCB4 (P3II-26, sc-58221) primary antibody was purchased from Santa Cruz Biotechnology, Inc. (Dallas, TX, USA), and ABCB1 (C219) primary antibody was purchased from Enzo Life Sciences, Inc (Lausen, Switzerland). Secondary anti-mouse IgG antibody was obtained from ThermoFisher (# 62-6520). The BCA protein assay kit was obtained from Thermo Scientific (Rockford, IL, USA).

### 4.2. Cell Lines and Culture Conditions

Madin–Darby canine kidney II (MDCKII) wildtype cells were obtained from the European Collection of Authenticated Cell Cultures (ECACC catalogue no. 00062107). A MDCKII-Abcb1 biallelic KO (Abcb1KO-MDCKII) cell clone was generated by GenScript (Leiden, The Netherlands) using the CRISPR/Cas9 system to knockout the canine Abcb1 gene. To confirm gene editing, after clonal expansion of the cells, genomic DNA was extracted, and the target regions were amplified by PCR using primer pairs for the expected mutation sites. The genotype of this KO clone was determined by Sanger sequencing by GenScript. The phenotype was verified through transport assays using the ABCB1/Abcb1-specific substrates digoxin and talinolol.

Sequence verified cDNA encoding human ABCB4 (NCBI Reference Sequence: NM_000443.3) was synthesized and cloned into pCDH-CMV-EF1-Puro using 5′ NheI and 3′ NotI by GenScript, and lentiviral particles were generated in HEK293FT cells (Invitrogen/ThermoFisher, Waltham, MA, USA). Transduced and antibiotic-selected Abcb1KO-MDCKII-ABCB4 cells were subjected to single cell cloning, and amplified clones were functionally tested for transporter-specific efflux activity. Four puromycin-resistant clones were isolated and tested for ABCB4 efflux activity using the known ABCB4 substrate digoxin [16] with a digoxin ER cut-off of two [93]. Based on this criterion and on ABCB4 mRNA and protein expression levels and cell culturing properties, the best clone was selected for continued validation and is hereafter referred to as the ABCB4 cell line. Empty vector transduced Abcb1KO-MDCKII-Mock cells were used as negative controls.

Cell cultures were maintained in Dulbecco’s modified Eagle’s medium (DMEM), 4500 mg/L of glucose, supplemented with GlutaMax^TM^, 10% *v/v* fetal bovine serum (FBS), 100 units/mL penicillin and 100 µg/mL streptomycin (all from Gibco/ThermoFisher) at 37 °C in 5% CO_2_ at 95% humidity. The medium was replaced three times per week. Cells were harvested using TrypLE™ Express (ThermoFisher) at 80 to 90% confluence and passaged or seeded.

For transport experiments, cells were seeded on Millicell^TM^ High pore density 0.4 μm PCF cell culture plate inserts (Millipore, Merck KGaA, Darmstadt, Germany) at a density of 2.27 × 10^5^/cm^2^ and grown for 6 days at 37 °C in an atmosphere of 5% CO_2_ and 95% relative humidity. The culture medium was changed once, the day before the experiment.

### 4.3. Calcein AM-Based Fluorescence-Activated Cell Sorting (FACS)

The cells were detached with TrypLE™ Express and washed with PBS once before use. The cells were suspended in 0.1 µM calcein AM diluted in DMEM without phenol red at a cell concentration of 1 × 10^6^ cells/mL. The cells were incubated with calcein AM at 37 °C for 30 min, protected from light, and cells in suspension were intermittently agitated. The cells were kept on ice until the flow cytometry measurements. Cellular fluorescence reflecting the transport activity was measured by an Attune Nxt cytometer (Thermo Fisher Scientific, Waltham, MA, USA) equipped with a blue (488 nm) laser. The calcein signal was detected in the BL1 channel (emission filter: 530/30 nm). Analysis of the data was carried out by the Attune Nxt Cytometer Software v3.1.2 (Thermo Fisher).

### 4.4. Bidirectional Transport Assays

Stock solution of the compounds were prepared freshly in dimethyl sulfoxide (DMSO). Donor solutions were prepared by diluting test compounds in Hanks’ balanced salt solution (HBSS) at pH 7.4. Prior to the transport studies, the cell monolayers were washed twice with pre-warmed HBSS and pre-incubated for 15 min in HBSS. The test compounds were added in triplicate to either the apical or basolateral sides of the monolayers, and HBSS was added to the receiver wells to start the transport assay. The cells were incubated with the compounds for indicated times at 37 °C, and permeability was measured both in the apical-to-basolateral (A-B) and basolateral-to-apical (B-A) directions. The apical chamber had a final volume of 0.125 mL, while the basolateral chambers contained 0.25 mL. Samples (35 µL) were withdrawn from the receiver compartments at 1 h, 2 h, 3 h, 4 h and 6 h, and from the donor compartments at 0 and 6 h. Sample volumes withdrawn at each time point were replaced with a corresponding volume of pre-warmed HBSS to maintain constant volumes during each experiment. All sample concentrations were corrected for dilution with replacement buffer during sampling.

In the inhibition studies, A-B and B-A permeabilities of digoxin (1 µM, traced with 0.17 µCi/mL ^3^H-digoxin) were investigated across ABCB4 cell monolayers in the absence or presence of increasing concentrations of putative inhibitors. Samples were collected at 3 h. To determine the amounts of radiolabeled substrates (including digoxin) transported, samples mixed with liquid scintillation cocktail were measured with a MicroBeta2 microplate counter (PerkinElmer).

### 4.5. LC-MS Sample Preparation and Analytics

Samples from the receiver compartments as well as donor compartments and dosing solution were diluted as necessary and injected into LC-MS/MS. The compounds were separated on a Zorbax RRHD Eclipse Plus C18 3 × 50 mm, 1.8 µm column (Agilent) using water and acetonitrile including 0.1% formic acid (*v*/*v*) as mobile phases in gradient modes. The HPLC system was coupled to an AB Sciex 5500 QTRAP Triple quadrupole linear ion trap mass spectrometer using an electrospray interface (AB Sciex LLC, Framingham, MA, USA). The mass spectrometer was operated in multiple reaction monitoring mode. Data analysis was performed with Analyst software (AB Sciex LLC).

### 4.6. Cell Monolayer Integrity

The integrity of tight junction dynamics in cell monolayers was monitored before and after the transport studies by measuring the transepithelial electrical resistance (TEER) using a World Precision Instrument, Epithelial Voltohmmeter system (Sarasota, FL, USA). The approximate TEER values were 100–120 Ω·cm^2^. Lucifer Yellow as a paracellular permeability marker was used to check that the highest concentration of the putative inhibitors did not disturb the integrity of cell monolayers. In brief, HBSS containing the putative inhibitors and Lucifer yellow (40 µg/mL) was added to the apical chambers. Samples were collected from the basolateral chambers. Fluorescence of the dosing and receiver samples was measured at an excitation wavelength of 450 nm and an emission wavelength of 520 nm with FLUOstar OPTIMA Microplate Reader (BMG Labtech, Germany). Monolayers with Papp values < 2 × 10^−6^ cm/s were considered intact.

### 4.7. Real-Time qPCR

Total RNA was extracted from cells using the TRIzol reagent according to the manufacturer’s instructions. Reverse transcription was performed using the Tetro kit, and TaqMan real-time qPCRs (assay ID Hs00240956_m1 for ABCB4) were performed (Invitrogen/ThermoFisher, Waltham, MA, USA). Canine Gapdh (Assay ID Cf04419463_gH) was used as an internal control. In each sample, ABCB4 mRNA was normalized to Gapdh mRNA. Relative gene expression data were given as the fold change (ΔΔC_T_). Each sample’s ABCB4 expression was first subtracted from its Gapdh expression to determine its ΔC_T_. The ΔC_TMock_ was then subtracted from the ΔC_TABCB4_ to determine the ΔΔC_T_. The relative expression of ABCB4 was determined by the formula 2^−(ΔΔCT)^.

### 4.8. Western Blot

Confluent Mock and ABCB4 cells grown on filters were washed twice with ice-cold PBS and lysed on ice in M-PER Mammalian Protein Extraction Reagent (Thermo Fisher, Waltham, MA, USA) buffer containing freshly added protease inhibitors. The lysate was centrifuged at 13,000× *g*, 4 °C for 10 min and the supernatant was collected. The total protein content was determined using the Pierce^TM^ BCA Protein Assay Kit (Thermo Fisher, Waltham, MA, USA). The lysate was mixed with LDS loading buffer 3:1, and total protein (30 μg/lane) was loaded and separated on 4–15% SDS-PAGE gradient gels. The proteins were then transferred to a polyvinylidene fluoride (PVDF) membrane, blocked with 5% non-fat dry milk for nonspecific binding and probed with primary antibodies specific to human ABCB4 (1:1000) or human ABCB1 (1:1000), respectively, overnight. Following three 5-min washes with TBST (Tris buffered saline + 0.1% Tween 20), the blots were probed with secondary antibody in TBST (1:2000, anti-mouse IgG-HRP) at room temperature for 1 h. After the washing steps, the blots were visualized by ECL reagent (Pierce™ ECL Western Blotting Substrate, #32106) by a BioRad ChemiDoc imaging system (Bio-Rad Laboratories, Inc., Watford, UK).

### 4.9. Calculations

All the experimental conditions were run in triplicate wells and repeated in three biological replicates. The apparent permeability coefficient (Papp, expressed in 10^−6^ cm/s) was determined from the amount of compound transported per unit time according to the following equation: (1)$$Papp=\left(\frac{\Delta Q}{\Delta t}\right)\times \left(\frac{1}{A\times C_{0}}\right)$$
where *ΔQ* (pmol) is the amount of substrate translocated to the receiver compartment by the end of incubation, *Δt* (s) is the duration of incubation, *A* (cm^2^) is the filter surface area, and *C*_0_ (pmol/cm^3^) is the initial donor concentration of the substrate.

Sink conditions were fulfilled.

The mass balance (recovery) was defined as the sum of the test compound recovered from the receiver compartment and the test compound remaining in the donor compartment at the end of the experiment, divided by the initial donor amount. This was calculated according to: Recovery (%) = (C_D(Fin)_V_D_ + C_R(Fin)_V_R_)100/(C_D(0)_V_D_), where C_D_ and C_R_ are the concentrations on the donor (D) and receiver (R) sides of the monolayer at the start (0) or end (Fin) of the experiment, and V is used for each of the respective volumes.

In all experiments, the recovery for all tested substrates was >70%

The efflux ratio (ER) was calculated as Papp_, B-A_/Papp_, A-B_.

The ER of digoxin as a probe substrate was determined at all concentrations of the inhibitor and in the presence of the vehicle only. IC_50_ values were calculated from ER values by nonlinear regression analysis in GraphPad Prism 9 (GraphPad, La Jolla, CA, USA).

## Figures and Tables

**Figure 1 ijms-24-04459-f001:**
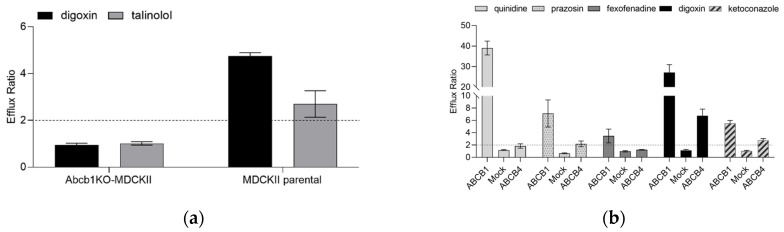
Efflux ratios for prototypic ABCB1 substrates. (**a**) Efflux ratios for digoxin and talinolol (1 µM, traced with 0.17 µCi/mL ^3^H-labelled probe) in the MDCKII parental and in the Abcb1KO cells. The dashed line indicates an efflux ratio of 2. Data are presented as mean ± standard deviation for 3 experiments each performed in triplicate. (**b**) Functional characterization of ABCB1, Mock and ABCB4 cells using prototypical ABCB1 substrates (1 µM, traced with 0.17 µCi/mL ^3^H-labelled probe). The dashed line indicates an efflux ratio of 2. Data are presented as mean ± standard deviation for 3 experiments each performed in triplicate.

**Figure 2 ijms-24-04459-f002:**
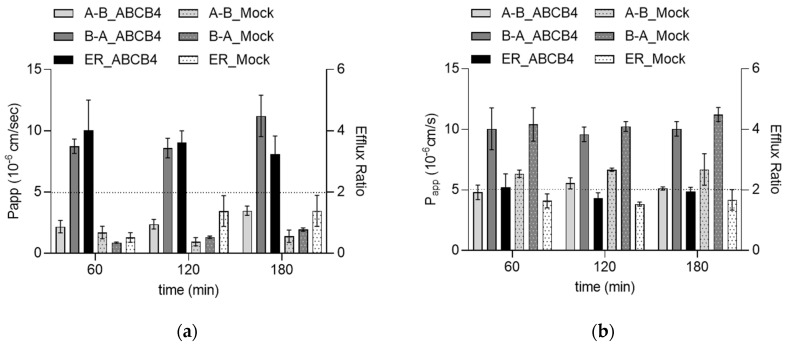
Transcellular transport of TKIs in the ABCB4 and Mock cells. (**a**) Bidirectional transport of gefitinib. ABCB4 and Mock cells were incubated with 0.5 µM gefitinib for 60, 120 or 180 min. Data are represented as means ± S.D. for three experiments. Gefitinib concentration in the samples was determined with LC-MS/MS. (**b**) Bidirectional transport of imatinib. ABCB4 and Mock cells were incubated with 0.5 µM imatinib for 60, 120 or 180 min. Imatinib concentration in the samples was determined with LC-MS/MS. Data are represented as means ± S.D. from three independent experiments.

## Data Availability

The data presented in this study are available on request from the corresponding author. The data are not publicly available due to the company participation.

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
