# Peer review of "A Unique In Vitro Assay to Investigate ABCB4 Transport Function"

_ijms, 2023, doi:10.3390/ijms24054459_

Round 1

Reviewer 1 Report

The manuscript describes preparation of a new cell line lacking ABCB1 transporter that they use for the expression of ABCB4 transporter and its characterization. Though the methodology and transport parameters for ABCB4 obtained with the new assay sound quite solid, the amount of new information provided in the study about the transporter function and regulation is quite low. In comparison with published data, the efficiency and usefulness of the new assay seems to be not too much advantageous. The data obtained with new proposed assay are only partially in line with previously published data. Also, I have several points to be addressed in the manuscript that should be improved.

Major points:

1. All results concerning characterization of new cell line in 2.1 (FACS, real-time PCR, protein expression) should be shown as at least supplementary material.

2. Time-course of digoxin transport (described in lines 152-158) should be shown at least as supplementary results.

3. Fig. 1B is not very clear as there is a big difference between the activity of ABCB1 for quinidine and digoxin than for all other substrates and in comparison with control mock cells and cells expressing ABCB4. I would suggest to show y scale interrupted and enlarge the scale of efflux ratio between 0-20, then also the cut-off value can be marked as in 1A. Concentrations of particular compounds used in the assay should be provided.

4. The content and length of Discussion section should be revised. In my opinion, it provides too much unrelated information, particularly in the first half. Some of this information can be moved to the Introduction section. Results presented in the study are discussed only in the last three paragraphs before conclusion.

Minor points:

1. line 79 – please, check the name of azole

2. List of oligonucleotides used for the construction of cell line should be provided as a supplementary table.

Reviewer 2 Report

ABCB4 is a phosphatidyl-choline flippase expressed in the canalicular membrane of hepatocytes and involved in the protection of the liver from the detergent effect of bile salts. ABCB4 and ABCB1 possess significant structural and sequence similarity resulting in partially overlapping substrate spectra of the two transporters. Inhibition of ABCB4 by certain drugs used for the treatment of various diseases may induce liver injury (DILI). Because of the overlapping substrate spectra of ABCB1 and ABCB4 polarized epithelial cells exclusively expressing ABCB4 could serve a convenient in vitro experimental model. Therefore, the Authors have generated an ABCB4-expressing Abcb1-knock out MDCKII cell line for transcellular transport assays.

The Introduction part is very long and not straightforward.  It should be shortened and reorganized.

In the Results section numerous experimental results are mentioned without demonstration by Figures or without showing them in a Supplementary dataset. Please show the results corresponding to the characterization of biallelic Abcb1KO-MDCKII cells: e.g., mRNA expression analysis, Western blot and calcein-AM assay. Similarly, please demonstrate ABCB4 expression both on mRNA and protein levels.

The results described in section 2.4 are also not represented in figures.

The Discussion section should be reformulated and shortened significantly. Please describe the strength and potential weaknesses of your assay system. What are the advantages/disadvantages of using a kidney distal tubule cell line instead of hepatocytes?  Compare your results with previous literature data, shed light on the potential causes of the differences.

Reviewer 3 Report

This very interesting manuscript characterizes a new cell model to study ABCB4- mediated transport. This report constitutes an important advance in the field. However, relevant data and justifications should be presented and some parts of the manuscript should be reorganized in order to improve the paper.

Major points:

-Advantages of this new cell model over other well established and widely used cell models such as LLCPK1 (with very low endogenous Pgp expression) should be discussed.

-Lines 138-138: Western blot images should be presented in order to validate the model.

-Lines 153-160: digoxin transport data should be represented in a graph (BA, AB, RE) to assess the validation of the model. Therefore, rationale of the use of digoxin at 1 um (line 175) should be justified.

-Some parts of the results should be moved to the discussion (lines 182-196, 199-202)

-Interpretation of Figure 2 is missing in the text.

-Lines 232-233: “we also tested whether the two most potent inhibitors, gefitinib and imatinib are ABCB4 substrates” Gefitinib and imatinib (IC50 0.81uM and 1.24 uM, repectively Table1) are not the most potent inhibitors, since there are other drugs with lower IC50 in Table 1 (cyclosporine IC50 0.46 uM, ivermectin IC50 0.39 uM, verapamil IC50 0.39 uM, valspodar IC50 0.15 uM).     

-Lines 233-239: transport data of gefitinib and imatinib should be represented (BA, AB, RE) for ABCB4 cells and mock to assess the validation of the model with a justification of the concentrations chosen.

-Lines 242-267: this part of the discussion should be removed due to lack of relevance for the paper.

Minor points:

-Line 169: “predicitive”

-Lines 488-493: mass balance and recovery were calculated but results are not presented.

-Testing ABCB4-mediated transport of phospatidilcoline would be very useful to validate the model. Why authors did not check this?

Round 2

Reviewer 1 Report

In my opinion, authors improved sufficiently the manuscript according to reviewer's suggestions. 

I have only one minor point:

Line 99-100 expression can be improved to: ....suited to identify potential drugs highly specific for ABCB4….

Author Response

Dear Reviewer,

We would like to thank you for taking the necessary time and effort to review the manuscript. Thank you very much again for your valuable and insightful comments that led to major improvements in the final version.

Remark: “Line 99-100 expression can be improved to: ....suited to identify potential drugs highly specific for ABCB4….”

Response: Thanks for your kind reminders. We revised the sentence as follows:

Line 99-100: We showed that this assay system is also well suited to identify potential drugs highly specific for ABCB4 by excluding overlapping specificity from Abcb1.”

Sincerely,

Zsuzsanna Gáborik, PhD

Reviewer 2 Report

According to my opinion the current form of the manuscript is appropriate for publication in IJMS.

Author Response

Dear Reviewer,

We would like to thank you for taking the necessary time and effort to review the manuscript. We sincerely appreciate all your valuable and insightful comments and suggestions, which helped us in improving the quality of the manuscript.

Sincerely,

Zsuzsanna Gáborik, PhD

Reviewer 3 Report

Most of the questions were adequately addressed.

Author Response

Dear Reviewer,

We would like to thank you for taking the necessary time and effort to review the manuscript. We sincerely appreciate all your valuable comments and suggestions, which helped us in improving the quality of the manuscript.

Sincerely,

Zsuzsanna Gáborik, PhD
